# Molecular Dynamics Simulation of Silicone Oil Polymerization from Combined QM/MM Modeling

**DOI:** 10.3390/polym16121755

**Published:** 2024-06-20

**Authors:** Pascal Puhlmann, Dirk Zahn

**Affiliations:** Lehrstuhl für Theoretische Chemie/Computer Chemie Centrum, Friedrich-Alexander Universität Erlangen-Nürnberg, Nägelsbachstraße 25, 91052 Erlangen, Germany; pascal.puhlmann@fau.de

**Keywords:** silicone, molecular dynamics, polymerization mechanisms

## Abstract

We outline a molecular simulation protocol for elucidating the formation of silicone oil from trimethlyl- and dimethlysilanediole precursor mixtures. While the fundamental condensation reactions are effectively described by quantum mechanical calculations, this is combined with molecular mechanics models in order to assess the extended relaxation processes. Within a small series of different precursor mixtures used as starting points, we demonstrate the evolution of the curing degree and heat formation in the course of polymer chain growth. Despite the increasing complexity of the amorphous agglomerate of polymer chains, our approach shows an appealing performance for tackling both elastic and viscous relaxation. Indeed, the finally obtained polymer systems feature 99% curing and thus offer realistic insights into the growth mechanisms of coexisting/competing polymer strands.

## 1. Introduction

Silicone is widely used in both industry and everyday life as it reflects inexpensive, non-toxic, and quite versatile compounds, with applications ranging from lubricants to sealants [1]. While the properties of solid silicones are tailored by the degree of crosslinking within polymer networks, silicone oils feature finite polymer strands, which, in the most simple case, are given by linear chains of methylated siloxanes, i.e., polydimethylsiloxane (PDMS), which is also named dimeticon. This compound is considered relatively environment friendly as it predominantly degrades via abiotic routes, namely fragmentation into smaller oligomers and final decomposition into silica, water, and carbon dioxide [2,3,4]. PDMS is widely used as lubricants/antifoam additives in both mechanical systems and cosmetics, and its global production amounts to hundreds of megatons per year [5,6,7,8,9,10].

To tune the properties of PDMS fluids, tailoring the chain length distribution is of central importance. For industrial purposes, the controlled syntheses of PDMS as well as the profound characterization of the final product properties were established from extensive empirical evidence collected many decades ago [1]. To flank these efforts, recent progress in experimental and theoretical characterization techniques offers molecular-scale insights. To this end, molecular simulation approaches appear to be particularly promising, as the desired atomic resolution is directly at hand. Pioneering studies were based on pre-defined PDMS strands and succeeded in reasonably reproducing a series of properties (both static and dynamical) of the bulk fluid [11,12,13,14,15,16]. Moreover, the perspectives of the molecular-scale insights available were nicely illustrated by assessing the migration mechanisms of guest species such as He and CH_4_ gas molecules [17].

While the characterization of a given silicone product is comparably straightforward to molecular mechanics simulations, modeling the actual polymerization reactions is more challenging. To go beyond stochastically connecting nearby precursor units, quantum chemical calculations are needed to account for bond formation/dissociation energies [18]. So far, only a small series of combined QM/MM or reactive MM (REAXFF) studies of silicone polymerization have been reported [19,20,21,22,23]. On one hand, these methods differ in terms of their accuracy in describing the interaction energy of specific configurations. On the other hand, a potentially even more crucial issue is related to the manifold of structures and network types. Indeed, the simulation methods differ in terms of how relaxation processes are implemented in the course of polymer growth. Proper relaxation becomes increasingly important with the overall progress of the polymerization process (and thus the decline in the available reaction partners). For this reason, the net turnover (i.e., the degree of curing) may be taken as a benchmark for the modeling protocols [24]. While early simulation studies often reached partial curing, leaving 10–30% of precursors unreacted, in the past decade, new approaches have emerged that achieve the curing degree of their experimental/industrial counterparts, which typically becomes very close to the full conversion of reactants [24,25].

In the following sections, we outline an efficient simulation protocol that uses a minimum of QM calculations for bond breaking/formation steps and employs MM-based annealing runs to cope with the relaxation processes. On the one hand, the tailoring of the PDMS length is elucidated by the example of trimethylsilanol (TMS) and dimethylsilanediol (DMSD) precursor mixtures. On the other hand, we inspect kinetic aspects by benchmarking the temperature dependence of our model simulations.

## 2. Methods

The formation of PDMS originates from condensation reactions linking monomer units from the precursor mixture or connecting already formed oligomers during the later stages of the polymerization process. The fundamental reaction, as illustrated in Figure 1 is readily assessable for QM characterization. All condensation reaction events are considered in a subsequent manner, using the dimerization of TMS in the gas phase as a QM reference. In turn, PDMS molecules, which are not involved in the specified Si-O-Si bond formation, are considered by MM methods. Likewise, for the reaction of PDMS oligomers, we treat the dimethyhlsilanediol/trimethylsilanol moieties that are not directly connected with the involved Si-OH moieties via MM models. This computationally very efficient approach thus reduces the QM part of our protocol to a single gas-phase reaction, whereas all interactions with the local environment and strain relaxation of the forming polymer strands are elucidated from MM [24]. To this end, our approach considers interactions between the QM subsystem and the MM part, but not vice versa. This leads to an approximate, non-hybrid QM/MM ansatz that reduces the QM calculations to an effective Δ*E_QM_* term computed only once for the reference system [24]. To assess the reaction heat, Δ*H*, we therefore assume the following:(1)$$\begin{array}{c} \Delta H=\Delta E_{QM}\left[{\left(CH_{3}\right)}_{3}SiOH+HOSi{\left(CH_{3}\right)}_{3}\to {\left(CH_{3}\right)}_{3}SiOSi{\left(CH_{3}\right)}_{3}\right]\\ \;-\Delta E_{MM}\left[{\left(CH_{3}\right)}_{3}SiOH+HOSi{\left(CH_{3}\right)}_{3}\to {\left(CH_{3}\right)}_{3}SiOSi{\left(CH_{3}\right)}_{3}\right]\\ \;+\Delta E_{MM}\left[all\;other\;atoms\right] \end{array}$$
where the Δ*E* terms refer to the changes in potential energy, respectively. The reacting subsystem is hence devised into two R/R’–(CH_3_)_2_SIOH moieties that are treated as R = R’ = CH_3_ in the QM calculations, whereas the MM models are employed to describe the underlying silicone chain fragments, R and R’, in general terms. 

To facilitate the modeling of relaxation processes, we directly anticipate water segregation from the silicone phase (Figure 1). Thus, all product water is immediately removed from our simulation models. To calculate Δ*E_MM_*, we simply switch between the underlying force fields and perform a short-term molecular dynamics (MDs) run to allow the Si-O-Si bond distances to relax. For complex reactions that involve large-scale reorganizations, this can be implemented via smooth topology transformations, as described in ref. [24]. 

This is followed by the polymerization of PDMS by condensation reactions. In our simulation protocol, the gas-phase reaction for k = m = 1 (and imposing vaporization of the formed water molecule) is used as the QM subsystem, whereas all other degrees of freedom are treated from MM calculations. 

## 3. Simulation Details

The quantum mechanical (QM) calculations were performed with the Gaussian 16 package [26] at the B3LYP/6-311G(2df) level. As starting points, we used pre-relaxed molecular arrangements that were prepared with the Avogadro package using the internal MMFF94 force field [27,28]. In turn, tailor-made interaction potentials of better accuracy (see also MM benchmarking in the results part) were adopted from ref. [29] for the MM calculations of the production runs.

For the MD simulations, the LAMMPS package (March 2023 version) [30] was used with a simulation time step of 1 fs. Constant pressure of 1 bar and a temperature of 300 K were implemented by the Nosè–Hoover algorithm, using relaxation constants of 1 ps and 0.1 ps, respectively [31,32]. The Si-O and Si-OH interaction models are based on pairwise potentials (no angular terms), as described by a combination of Coulomb and Buckingham terms (short-range repulsion and London interactions) [29]. This setup features, in principle, reversible formation of Si-O bonds, including the possibility of PDMS rupture and re-formation (including cyclization of PDMS chains upon mechanical loading). In turn, the Si-CH_3_ and H_3_C-Si-CH_3_ bond and angle potentials, respectively, were taken from the force field of Smith et al. [15]. In our models, the Si-C bond is hence assumed to be permanent, which appears reasonable, since tensile stress would mainly apply to the backbone of PDMS and much less to the lateral –CH_3_ groups.

For the evaluation of the pairwise potentials, a real-space cutoff distance of 12 Å was applied. This implies full consideration of all interaction terms when discussing the benchmark clusters modeled in the gas phase. In turn, for the condensed phase models, particle mesh Ewald summation was used to describe the long-range electrostatics [33]. The MD simulations of the polymerization runs all start from entirely unreacted precursor mixtures, modeled as bulk liquid phases. For this, cubic simulation cells featuring 500 molecules were subjected to periodic boundary conditions. The initially random arrangement of precursor molecules was equilibrated according to the underlying liquid state from 2 ns relaxation runs at 300 K and 1 atm, respectively. In parallel setups, the polymerization of three precursor liquid models was investigated, namely 500:0, 450:50, and 400:100 mixtures of DMSD/TMS species, respectively. Molecular visualization and analyses were performed with VMD [34] and Python 3.10.

## 4. Results

The quantum mechanical calculations are focused on the fundamental reaction indicated by Figure 1. Thus, geometry optimization of the stand-alone TMS, the dimerization product, and a water molecule at infinite distance were performed. To this end, the gas-phase QM calculation already reflects water dissociation from the polymer, and water segregation is considered implicitly. This boosts the modeling of relaxation processes during the MM simulations considerably. The effective QM/MM correction term for the QM subsystem reduces to a constant of Δ*E_QM_* = −0.023 eV, whereas the actual reaction heat of the total reaction strongly depends on the MM terms in Equation (1).

Since the dimerization of TMS is studied in both QM and MM, we suggest this as a simple benchmark for our MM models. In Figure 2, we show the minimum energy configurations as obtained from both approaches. The overall root-mean-square deviation amounts to less than 0.3 Å. In turn, the Si-O and Si-Si distances are obtained from the MM model as 1.57 and 3.15 Å, respectively, and describe the QM reference (1.64 and 3.25 Å) with even better accuracy.

For modeling the polymerization process, our MD simulation protocol describing the polymerization progress is based on an iterative procedure [24]. The starting system is first propagated for 1 ps. We then inspect the distances of nearby silanol moieties and select the nearest OH∙∙O contact as a candidate for the condensation reaction. To evaluate Δ*E_MM_*, we must sample the system twice, using the MM model of (i) the reactant state and (ii) the possible product of a polymerization step. The latter is introduced by simply switching to the corresponding force field and relaxing the new Si-O-Si bonds in a short MD run. This is typically observed within much less than 1 ps; however, we use 1 ps to ensure proper formation of the Si-O-Si bridge. Upon modifying the simulation system in this manner, local interactions change significantly. We therefore use a smaller integration time step of 0.1 fs for these relaxation runs.

Next, sampling of the time averages is required to provide the MM energy before and after the linking reaction. In our previous study of epoxy resin formation, this was obtained from ~5 ps scale runs [24]; however, for the present system, we found 10 ps relaxation more appropriate to discriminate Δ*E_MM_* at reasonable accuracy. Upon exothermic reaction energy, the reaction attempt is accepted, and the iteration continues. In turn, we dismiss reaction attempts that would have led to endothermic energy change.

On this basis, our simulation models are propagated upon every reaction attempt, irrespective of whether the condensation reaction is successful or not. For this reason, our MD runs describe the relaxation of the forming polymer in an adaptive manner, providing extra time where needed for annealing of the overall system.

To benchmark the overall polymerization protocol, it is educational to compare our QM/MM results to QM reference calculations. While the dimerization of TMS is used as an input parameter in Equation (1), we calculated the formation energy in the gas phase of a series of oligomers featuring up to eight Si-O-Si bridges. From this, we found that the deviation in the formation energy from explicit QM and the QM/MM approximation was only 2%.

During the initial stages of polymerization, an abundance of nearby Si-OH moieties is available, and nearly all reaction attempts are exothermic. However, at later stages, the polymer strands become partially ‘jammed’, and increasing viscosity calls for longer relaxation times. To monitor this, we inspect the profiles of the acceptance probability of the individual polymerization steps as a function of the overall curing degree η (Figure 3, left). Up to curing of η = 80–90%, we find relatively quick conversion of the precursors and relaxation from 10 to 100 ps scale MD runs suffice to find new reaction partners.

However, the last 10% are much more difficult to achieve, as, inevitably, the acceptance probability converges to zero. While some limitations of the number of attempted reactions are indispensable to all approaches to modeling polymerization, our method benefits from the particularly inexpensive combination of QM and MM techniques. As a consequence, we can afford to set the upper boundaries quite generously. By imposing a maximum of 10^5^ failed reaction attempts as the convergence criterion, we indeed achieve η ≈ 99%. To point out the challenges encountered upon reaching near-perfect curing of the polymer in Figure 3 (right), we also show the evolution of the curing degree η as a function of the log10 of the number of total reaction attempts needed.

A typical example of the evolution of our simulation models is illustrated in Figure 4. Starting from a pure liquid of the precursors (here, 500 DMSD molecules), we find dimer, trimer, etc., formation without notable spatial preference. However, with increasing progress of the polymerization, here η ≥ 50%, the reservoir of unreacted DMSD is depleted, and polymerization instead occurs via reactions of oligomers. Since oligomer mobility reduces with increasing chain length, it is quite intuitive that the relaxation times needed between successful growth steps also increase. By contrasting analogous PDMS formation runs performed at different temperatures, we indeed find changes in the profiles of the number of attempts needed to propagate polymerization beyond curing of η > 90% (Figure 3).

Upon polymerization (and removal of the product water), the simulation systems experience considerable volume change. Indeed, the cubic cells of 43.7, 44.0, and 44.3 Å dimensions of the 500:0, 450:50, and 400:100 DMSD:TMS precursors, respectively, were reduced to 39.0, 39.5, and 40.0 Å dimensions during the corresponding curing processes, respectively.

Likewise, the process temperature also affects the distribution of the length of PDMS chains in the final product. In Figure 5, we show occurrence profiles of the PDMS chain lengths as functions of (a) the curing degree and (b) the different temperature regimes (250, 300, and 350 K) investigated. The corresponding analyses of oligomer specie refers to PDMS, resulting from the pure DMSD precursor liquid. Drastic effects in the chain length distribution of the finally obtained silicone oils are, of course, achieved by (c) modifying the composition of the precursor setup. This is illustrated by contrasting a series of polymerization runs (300 K) in which we substitute 10% and 20% of the DMSD with TMS (Figure 5c). While DMSD offers two OH groups for condensation reactions, TMS features only one and thus acts as a terminator for chain growth. Indeed, even the low percentages investigated in these two case studies already lead to a dramatic reduction in the PDMS chain length.

For this series of polymerization runs, we also monitored the heat of reaction as a function of the curing degree. Figure 6 shows the evolution of the heat released in the individual reaction steps, as characterized during our iterative polymerization scheme. Comparing the simulations performed for pure DMSD and analogous runs, but substituting 10 and 20% of the DMSD by TMS, we find that the reaction heat per polymer growth step is roughly constant for 0 ≤ η ≤ 70%. On the other hand, the reaction heat (for the pure DMSD precursor system) reduces by about 0.10 eV upon approaching full curing (investigated up to η = 99%). Based on our simulations, we thus identify a gradual increase in steric hindering that slows polymer growth both from a kinetic viewpoint (increase in viscosity) and energetically (residual strain in the PDMS chains).

These observations nicely comply with the experimental evidence collected for gas-phase reactions of TMS dimerization and the overall heat of PDMS formation. The former refers to η = 0%, and the experimentally observed reaction enthalpy of 0.22 eV [35] is in reasonable agreement with the 0.26 eV result estimated from our simulations (Figure 6). In turn, the experimental values for the heat of bulk PDMS polymerization, thus referring to the integral for η = 0 to ~100%, are within the range of 0.17 to 0.21 eV per monomeric unit, respectively [36]. In turn, by averaging the overall heat contributions from each reaction step (η = 0 to 99%), our simulations indicate a bulk PDMS heat formation of 0.24 eV per monomeric unit. Accordingly, the strain energy per monomeric unit is estimated to be 0.02 eV from our simulations, whilst the experiments indicate 0.01–0.04 eV, respectively.

The final models of silicone oil as obtained from the three types of precursor compositions (0, 10, and 20% TMS; curing at 300 K) were subjected to additional 10 ns MD runs to characterize the bulk liquid properties. While the overall density (1.04, 1.01, and 0.95 g/cm^3^) reduces with increasing TMS content, the Si-O radial distance distribution functions only show subtle differences (Figure 7). On the other hand, the dynamics of the different silicon oil formulations display a strong dependence of molecular mobility on the average PDMS chain length. Indeed, for 0/10/20% TMS, we find *ν* = 65/15/9 and diffusion constants of 0.3/0.6/2.3∙10^−7^ cm^2^s^−1^, respectively (Figure 7).

## 5. Conclusions

Our simulation protocol, which was originally developed for epoxy resins, was successfully transferred to the investigation of silicone polymerization. For this, a small set of quantum chemical calculations remain indispensable, whilst the main part of the simulation could be performed with molecular mechanics models. Thanks to the efficiency achieved in computational performance, we are able to characterize the polymerization of up to 10 nm scale simulation systems featuring hundreds of reaction precursors. While the maximum length of PDMS strands is intrinsically limited, the investigated simulation cells are sufficiently large enough to avoid the artificial connection of polymer chains via periodic boundaries. Thus, finite oligomers are prepared, and the resulting model qualifies for characterizing silicone oil from MD simulation. To this end, future directions involve the investigation of aging from thermal or mechanical impacts, such as the formation of cyclic PDMS, which was found only in trace amounts (~1%) in the present study. Moreover, it is straightforward to consider tri- or quarto valence silicone linkers as needed for the formulation of solid silicones.

## Figures and Tables

**Figure 1 polymers-16-01755-f001:**
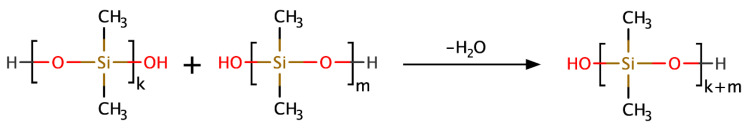
Polymerization of PDMS by condensation reactions. In our simulation protocol, the gas phase reaction for k = m = 1 (and imposing vaporization of the formed water molecule) is used as the QM subsystem, whereas all other degrees of freedom are treated from MM calculations.

**Figure 2 polymers-16-01755-f002:**
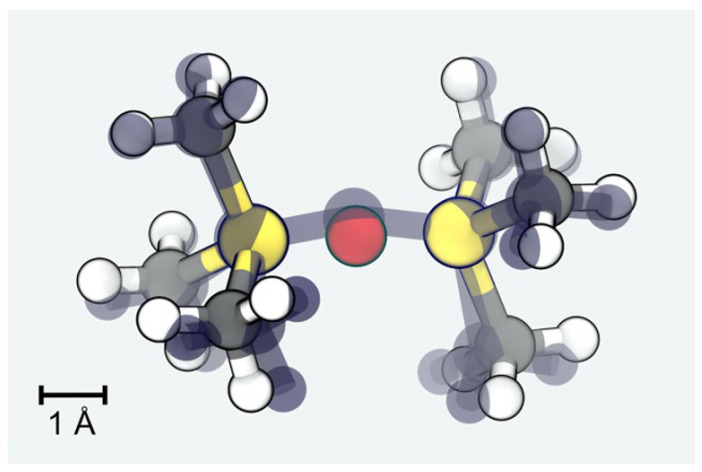
Comparison of the structure of hexamethyldisilane as obtained from dimerization of TMS. The atomic positions resulting from the MM model are shown as solid spheres (Si: yellow; O: red; C: gray; H: white), whereas the QM structure is indicated by a transparent ball and stick representation.

**Figure 3 polymers-16-01755-f003:**
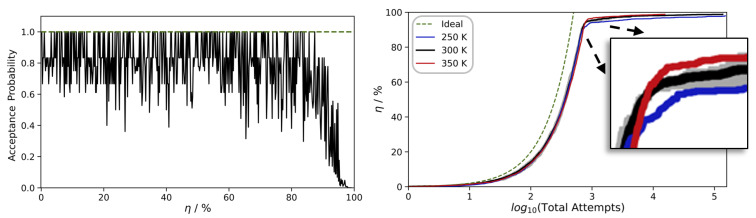
Interplay of curing degree and relaxation times needed as observed for a typical polymerization run. Left: evolution of the acceptance probability of the individual polymerization steps as a function of η. Right: curing degree as a function of the cumulative sum of reaction attempts needed. For comparison, the dashed curves shown in green indicate an idealized scenario in which all reaction attempts are successful. The solid curves shown in blue, black, and red refer to averages taken from three independent runs each performed at 250, 300, and 350 K, respectively, whereas the root-mean-square deviations are indicated by transparent color representations.

**Figure 4 polymers-16-01755-f004:**
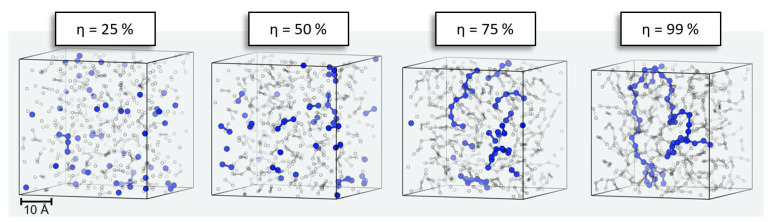
Curing of PDMS at 300 K and 1 atm using 3D periodic simulation cells. Starting from 500 DMSD precursor species, curing up to η = 99% is achieved, leading to a density increase from 0.91 g/cm^3^ to 1.04 g/cm^3^, respectively. Only Si atoms are shown. In the final system, a siloxane chain of 54 DMSD building blocks is highlighted in blue. To illustrate the formation process via reactions of monomers and oligomers, the same coloring is applied to earlier stages of polymerization.

**Figure 5 polymers-16-01755-f005:**
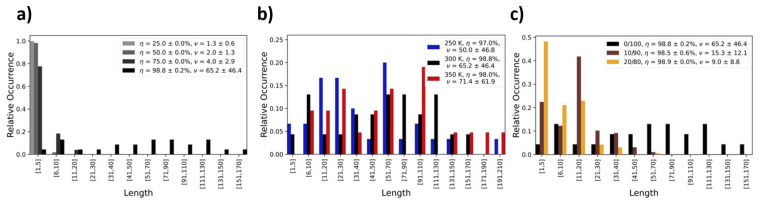
Distribution of PDMS chain length as functions of (**a**) curing degree, (**b**) process temperature, and (**c**) precursor composition. The mean chain length (ν) is averaged from three independent polymerization runs. From the root-mean-square deviations we assess the polydispersity index (PDI = M_w_/M_n_) as PDI = 1.73/1.49/1.59 and PDI = 1.49/1.60/1.91, as functions of temperature (T = 250/300/350 K and 100% DMSD precursors) and composition (0/10/20% TMS and T = 300 K), respectively.

**Figure 6 polymers-16-01755-f006:**
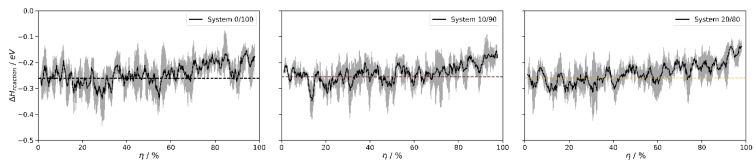
Heat of individual reaction steps as observed during the curing of PDMS. Data are shown for various DMSD/TMS precursor mixtures, all cured at 300 K and 1 atm, respectively. The curves shown in black refer to averages taken from three independent runs, whereas the root-mean-square deviations are indicated in gray. Left to right: profiles sampled for 0/10/20% TMS content, respectively, showing rough constant reaction heat (<ΔH>_0<η<70%_ = 0.26/0.25/0.26 eV, as indicated by the dashed lines) up to η < 70%. Upon curing by 99%, the reaction heat is gradually reduced by about 0.1 eV because of the mechanical strain within the (amorphous) agglomerates of PDMS chains. The overall heat of formation is sampled as <ΔH>_0<η<99%_ = 0.24/0.24/0.24 eV per monomeric unit for the silicone oils prepared from DMSD/TMS precursor mixtures of 0/10/20% TMS content, respectively.

**Figure 7 polymers-16-01755-f007:**
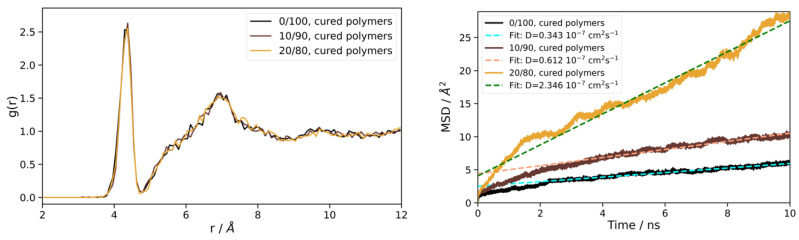
Characterization of the finally obtained silicone oil models from curing the precursor systems of 0/10/20% TMS content at 300 K and 1 atm. Left: radial distribution functions of the Si-O distances. Right: mean-squared-deviation profiles as function of time as used for assessing the diffusion constants. For the latter, linear fits are applied to the last 7 ns, as indicated by the dashed lines.

## Data Availability

The original contributions presented in the study are included in the article, further inquiries can be directed to the corresponding authors.

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
