# Peer review of "Molecular Dynamics Simulation of Silicone Oil Polymerization from Combined QM/MM Modeling"

_polymers, 2024, doi:10.3390/polym16121755_

Round 1
Reviewer 1 Report
Comments and Suggestions for Authors
This manuscript outlines a study on molecular dynamics simulation of silicone oil polymerization from combined QM/MM modeling. The findings are positive and provide valuable recommendations despite the increasing complexity of the forming bundle of polymer strings, as well as appealing performance for tackling both elastic and viscous relaxation. To enhance the manuscript's persuasiveness, some key areas require improvement before it can be considered for acceptance by the editor.
1. The introduction section lacks sufficient information about previous research.
2. Multiple citations [15–10]; [11–16]; and [19–23] for a single point should be avoided.
3. The equation description and the source of this content need to be articulated clearly.
4. The authors used with a simulation time step of 1 fs. I strongly suggest to the author that a femtosecond should be thoroughly cross-checked with appropriate references for accuracy.
5. The figures need significant improvement in quality, especially Figure 5.
Author Response
Thanks for the useful hints. A detailed list of changes is attached.

Reviewer 2 Report
Comments and Suggestions for Authors
The paper by Pascal Puhlmann and Dirk Zahn deals with a molecular simulation of the chemical reaction in silico. The authors use two complementary methods of quantum and molecular mechanics (QM/MM) to reproduce the reaction synthesis of the silicone oils trimethlylsilanol and dimethlysilanediol mixtures. This article may be of interest to readers involved in the design of QM/MM simulation schemes. However, in its current form there are a number of unclear points in the article, which would be of interest to the authors for clarification. My comments are given below.
1) Line 56. The authors write "trimethyl- and dimethylsilanediol (TMS and DMSD)". The full name of the first compound should be given, as it clearly contains only one OH group, it is probably trimethylsilanol.
2) Lines 66-67. The authors write "di/tri-methyl silanediol". As far as I know, all diols contain two OH groups. This is true for dimethylsilanediol. TMS contains only one OH group, so it is not correct to write trimethylsilanediol. It looks like trimethylsilanol. Authors should check the correct spelling of this compound in the text of the article.
3) The authors write that they considered the reaction (𝐶𝐻3)3𝑆𝑖𝑂𝐻+𝐻𝑂𝑆𝑖(𝐶𝐻3)3→(𝐶𝐻3)3𝑆𝑖𝑂𝑆𝑖(𝐶𝐻3)3 in the gas phase. If two monomers react, there is no strong objection. From the text of the article, I did not quite understand how the authors go into the gas phase and back when the end of the oligomeric chain reacts with the TMS and DMSD monomers, and when the two ends of the oligomeric chains react. This should be clearly stated in the methodological part of the article. That is, the question is: how is the linkage of the reacting comonomer to the oligomer chain taken into account in the gas phase? This point is lost in the description of the simulation protocol and should be explained. Perhaps Figs 1 and 2 should be updated to make the explanaton more clear.
4) When the authors describe the preparation of a mixture in a simulation cell, the size of the simulation cell and the initial density of the comonomer mixture should be specified.
5) When describing the gas phase reaction shown schematically in Figure 1, the authors write that they remove water. It is not entirely clear whether the authors are removing water from the simulation cell as new bonds are formed during the reaction? If so, the cell size should decrease as a result of MM modeling because the authors use a barostat. How do the initial and final sizes of the simulation cells compare? Is there enough relaxation time to equalize the density and pressure in the cell? How did the authors control for this? Possible changes in cell edge size should be shown in Figure 4. In this figure, it looks like the cell size is actually decreasing. This is also indicated by the change in density given in the signature.
6) In the caption to Figure 1, it should be noted that Me is a -CH3 group.
7) Figure 2. It is not entirely clear why only TMS dimerization was chosen for the example? Perhaps it is worth discussing how the addition of a new monomer affects the structure of the chain and how the relaxation time of the system changes?
Author Response

(The authors gave the same response as above.)

Round 2
Reviewer 2 Report
Comments and Suggestions for Authors
The authors have responded to most of my comments. However, there are still some unclear points in their manuscript that need to be corrected.
1) Abstract. The authors write,
“Despite the increasing complexity of the forming bundle of polymer strings, …”
Perhaps the authors meant "polymer chains" instead of "polymer strings". The "polymer strings" are the parts of a musical instrument that produce sound.
It is also not entirely clear what the authors mean by the word "bundle". Is it a synonym for the word "bundle"? The authors mention this word in line 263, where they mention "the bundles of PDMS chains". The authors do not mention that such structures can form inside the simulation cell anywhere else in the article. The presence of such aggregates means that the chains are mutually ordered. This makes the samples non-amorphous. If the authors actually observe the formation of bundles of polymer chains, this should be described. If the authors mean something else, it should be clarified.
2) Lines 55-56. The authors write,
“ While early simulation studies often reached partial curing leaving 10-30% of precursors unreacted, modern approaches should achieve the curing degree of their experimental/industrial counterparts – which typically gets very close to full conversion of reactants.”
Here I would like to contradict the authors, since it can be found in previously published articles that the conversion degree of 98% has already been achieved in the polymerization of epoxy networks, for example in the framework of the use of multiscale modeling, see doi:10.1021/ma502220k.
3) Lines 69. The authors write,
“All condensation steps”
It's not entirely clear what that means? Since "condensation steps" does not appear anywhere else in the text, this phrase (which looks like non-standard terminology) needs to be rewritten in a different, i.e. clearer, way.
4) Lines 74-75, Authors write,
“This computationally very efficient approach thus reduces the QM part of our protocol to a single gas phase reaction”
It is difficult to understand from the article how the authors perform the polymerization reaction? Although the authors write in the abstract that "we outline a molecular simulation protocol", it is not outlined anywhere in the article. This is the major deficiency of the article. Therefore, the authors need to present it step by step or in the form of a block diagram in the Methods section. When describing it, the authors should start by describing the construction of the system, then they have to indicate in which step of the calculation scheme the relaxation of the system takes place (apparently it is implemented within the framework of molecular mechanics), then in which step the chemical reaction takes place (apparently it is implemented within the framework of QM calculations), and so on. How often these steps are repeated, what is the criterion for stopping the protocol. Without this, the explanations in the article look unclear. Again, it is necessary to indicate what role Eq. 1 plays in the construction of the calculation scheme.
5) Lines 127-128. The authors write,
“The quantum mechanical calculations are focused on the fundamental reaction indicated by figure 1.”
It is not entirely clear to me, and I asked in a previous review, were all the bond formation reactions done in the gas phase (formation of a dimer, addition of a monomer to a chain, and formation of bonds between chains)? Or did the authors use QM to parameterize ReaxFF in Lampps? This should be described in the methodological part of the article and also mentioned in the description of the chemical reaction modeling protocol developed by the authors.
6) Lines 153-154. The authors write,
“This is typically observed within much less than 1 ps, however we use 1 153 ps to ensure proper formation of the Si-O-Si salt bridge.”
I don't quite understand why the authors refer to the oxygen-silicon backbone (Si-O-Si) as a "salt bridge"? The salt bridge is a type of the ionic bond! In the case of the reaction in consideration, covalent bonds are formed. Therefore, the term "salt bridge" is not correct in this context.
Author Response
Thanks for the comments&hints - a detailed list of changes is attached.

Round 3
Reviewer 2 Report
Comments and Suggestions for Authors
The authors have corrected all inaccuracies in the article. It's recommended for publication.